# Analysis of proposed carbon capture projects in the US power sector and co-location with environmental justice communities

Yukyan Lam[1]*, Jennifer Ventrella[1,2], Ana Isabel Baptista[1,2], Juan David Rodriguez[1]

1 Tishman Environment and Design Center, The New School, New York, New York, United States of America, 2 Milano School of Policy, Management and Environment, The New School, New York, New York, United States of America

* ylam.pub@gmail.com

**Data availability statement:** All relevant data are within the paper and its Supporting Information files.

**Funding:** The author(s) received no specific funding for this work.

## Abstract

In recent years, there has been a proliferation of new federal investments and policy support for "carbon management" technologies, such as carbon capture and storage (CCS), as a strategy to mitigate the United States' greenhouse gas emissions (GHGs). The equity implications of deploying these technologies–particularly their impacts on low-income communities and communities of color, or environmental justice (EJ) communities–have been understudied. A prominent example of this is seen in the US power sector, where CCS has been proposed as a means to mitigate the carbon dioxide emissions of fossil fuel-fired power plants, one of the major sources of GHGs in the country. EJ community leaders alongside some environmental organizations and researchers have voiced deep concerns about how CCS may exacerbate environmental injustice, given that it is itself input-intensive and can prolong the life of polluting fossil fuel infrastructure, which is disproportionately sited in low-income communities and communities of color. To begin to fill the gap in analyses of the equity implications of carbon management, we conducted a spatial analysis of CCS projects proposed for the power sector and their co-location with EJ communities. Compiling a proposed project list from four CCS databases, we found that 33 of the 35 projects were located in EJ communities, and that additionally, 423 of the 497 (or 85%) EJ census block groups located within three miles of at least one proposed project currently face heightened environmental stress. These results illustrate both the feasibility and the necessity of analyzing the co-location of proposed CCS buildout in EJ communities, and add to the nascent body of literature evaluating the impacts of carbon management technologies such as CCS on these communities.

**Competing interests:** The authors have declared that no competing interests exist.

## Introduction

Carbon capture and storage (CCS) or carbon capture, utilization, and storage (CCUS) refer to processes where $CO_2$ is captured and separated at the point of combustion and transported for use or storage. Proponents of CCS maintain that it can be used to abate the $CO_2$ emissions of the power sector either by capturing the $CO_2$ emissions of coal and natural gas power plants, or by capturing the $CO_2$ emitted to make fossil fuel-derived hydrogen as an alternative fuel source that power plants can co-fire [1–3]. However, fossil fuel infrastructure is disproportionately sited in low-income communities and communities of color [4–7], and representatives of these environmental justice (EJ) communities have voiced deep concerns about how expanding and perpetuating this infrastructure will compound the socio-environmental burdens they already face [8–10]. One of the primary concerns is the potential for increased co-pollutant emissions, such as particulate matter, sulfur dioxide, and nitrogen oxides, in EJ communities. Co-pollutant emissions emitted by fossil fuel-powered plants and other polluting sources can contribute to negative, localized health impacts in communities such as adverse birth outcomes, cardiovascular disease, respiratory impacts, and cancer [11–14]. The linkage between co-pollutant emissions and negative health impacts underlines the importance of assessing the co-location of energy infrastructure in EJ communities and examining carbon management proposals beyond their purported mitigation of greenhouse gas (GHG) emissions.

The Intergovernmental Panel on Climate Change's (IPCC's) discussion of carbon dioxide removal and carbon capture technologies reveals the prominence of these approaches in climate plans and policies [15]. Within the United States, although not entirely new, financial incentives and other policies advancing CCS have proliferated especially in recent years. For example, the 2021 Infrastructure Investment and Jobs Act (IIJA) and the 2022 Inflation Reduction Act (IRA) introduced significant expanded federal funding and incentives for CCS [16–18], and in 2024, the US Environmental Protection Agency (EPA) classified CCS as a best system of emissions reduction (BSER) for new natural gas and existing coal plants [19,20]. The federal government, through agencies like the Department of Energy, has assumed the role of a catalytic investor in so-called "carbon management" technologies including CCS, while its capacity to regulate carbon management activities has been de-emphasized, posing new risks to EJ communities and the broader public [21]. Carbon management is a term used to describe a suite of technologies that remove carbon dioxide from point sources and the atmosphere for permanent storage or use in industry [22]. In addition to CCS, several other examples of carbon management technologies include direct air capture, bioenergy with carbon capture and sequestration, hydrogen fuels, and renewable natural gas. Another major factor in the prominence assumed by CCS is the fact that climate modeling, which has formed a basis for some policymaking, invokes CCS in a way that assumes high rates of efficacy and neglects negative externalities and equity implications [21].

Research has shown that CCS has limited potential to reduce GHG emissions and may even increase them [23,24], and other researchers have documented known health and safety risks along the CCS supply chain including extraction, capture,

transport, and storage [10]. However, policymakers and other proponents of CCS have yet to conduct an environmental justice analysis of proposed CCS buildout in the power sector. While CCS investments cover additional sectors, such as industrial decarbonization and hydrogen as a transportation fuel, prominent national level debates about the use of CCS in power sector regulations motivated the focus of this study. This paper describes an illustrative analysis of the potential for CCS to exacerbate risks in EJ communities through a spatial analysis of proposed CCS projects at power plants.

## Methods

### Proposed CCS projects

A list of proposed CCS projects in the US power sector was compiled drawing from databases of the following sources: 1) DOE Energy Technology Laboratory (NETL), 2) Global CCS Institute, 3) International Energy Agency (IEA), and 4) Clean Air Task Force (CATF) (see S1 Table for a full list of projects included in the analysis and the access dates for each database used). Two additional facilities were included based on information provided by participants at the National Symposium on Climate Justice and Carbon Management at the Wingspread Center, Wisconsin from June 1–4, 2023. Latitude and longitude coordinates were provided for projects appearing in the DOE database. For projects not included in the DOE database, we estimated their location based on publicly available data about the power plant at which they would be installed or at an approximate location. Approximate locations were estimated for seven of the 35 plants. To perform the estimation, secondary research sources such as project announcements, reports, or other documentation were used to triangulate an approximate project location. Approximated locations are a potential source of error in the analysis, however, these are random, not systematic, errors, and therefore should not inherently bias the data towards or away from any particular population. Our inclusion criteria included any planned CCS facility in the power sector across all four databases. The application of this criteria resulted in a final list of 35 planned facilities included in the analysis (see S1 Appendix and S1 File). We did not include operational plants because at the time of the analysis the only operational CCS facilities in the power sector were at pilot/demo scale, not commercial scale.

### Spatial analysis of co-location with EJ communities

It was next determined whether each facility was located within three miles of an EJ community. The distance of three miles is consistent with other EJ co-location studies and is used in US EPA's Power Plants and Neighboring Communities mapping tool [25]. EJ communities were considered to be those census block groups (CBGs):

• Whose percentage of people of color is equal to or greater than the state's overall percentage people of color; or

• Whose percentage of population living at or below twice the federal poverty level is equal to or greater than the state's percentage of population living at or below twice the federal poverty level.

These race and income-based criteria were based on the Equitable and Just National Climate Forum (EJNCF)'s recommended criteria for defining EJ communities for purposes of targeting power sector emissions reductions [26]. The EJNCF is a group of national environmental organizations and environmental justice organizations dedicated to advancing a national climate and environmental policy agenda that centers on environmental justice. As articulated by EJNCF, using race and income-based criteria to define EJ communities is consistent with scientific literature showing those two factors to be key predictors of environmental inequality, as well as with federal and state government policy guidance on how to identify EJ areas [26,27].

Using these demographic criteria to examine the fenceline populations of the CCS facilities is also consistent with the EPA's Power Plants and Neighboring Communities mapping tool [28], which displays the "demographic index," an index that averages the percentages of low-income individuals and people of color (POC), as well as the low-income percentage, the POC percentage, and the linguistically isolated population percentages, in the 3-mile areas surrounding a power

plant. Rather than average the demographics over the 3-mile area to determine whether the EJ thresholds were met, we instead focused on whether the 3-mile area around a plant contained any EJ CBG in whole or in part. This latter option was considered more protective and inclusive, as it would allow for the inclusion of smaller EJ areas within the 3-mile radius.

The dataset of CBGs was obtained from EJSCREEN in July 2023, and the above criteria were applied to the demographic indicators contained in that dataset [29]. The demographic indicators in this vintage of EJSCREEN are from the American Community Survey, 5-year estimates for 2017–2021. Data cleaning and application of the criteria were done in R. The dataset of CBGs was joined to a shapefile of CBG boundaries for 2021, obtained from the US Census Bureau [30]. Spatial analysis and mapping were performed in QGIS.

The proximity analysis was conducted in QGIS (version 3.32) using the 'Buffer' tool to create 3-mile radius buffer zones around the points of interest. Input data included a point layer representing the facility locations, a shapefile layer representing CBGs, and a CSV layer that included socio-demographic data. The coordinate reference system (CRS) was set to ESRI 102003 to ensure accurate distance calculations in miles. Features within the buffer zones were identified and analyzed through spatial joins with the EJ CBG and supplemental indices datasets. Additional documentation for each tool can be found at the QGIS website.

### Analysis of social and environmental burden in EJ communities near proposed facilities

The social and environmental burden in those EJ CBGs located within three miles of a proposed CCS power plant project was examined by relying on indicators of burden already contained in the EJSCREEN dataset. Specifically, EJSCREEN contains "supplemental indices," which combine a five-factor demographic index (low income, unemployment, limited English, less than high school education, and low life expectancy) with each one of 13 environmental indicators (PM2.5, ozone, diesel, air toxics cancer risk, air toxics respiratory hazard index, toxic releases to air, traffic proximity, lead paint, proximity to a Risk Management Plan facility, proximity to a facility managing hazardous waste, Superfund proximity, underground storage tanks, and wastewater discharge). For each CBG, EJSCREEN has an indicator which indicates the number of supplemental indices exceeding the 80th percentile relative to the rest of the country, and a similar indicator indicating the number of supplemental indices exceeding the 80th percentile relative to the rest of the state.

We tallied the number of EJ CBGs within three miles of an implicated facility that had at least one supplemental index exceeding the 80th percentile. We initially used the national-level percentiles, but later established that the analysis yielded the same result when using the state-level percentiles in this vintage of EJSCREEN. In order to ensure that CBGs located near more than one implicated facility were not double-counted, we merged three-mile buffers around the implicated facilities, intersected the buffers with the EJ CBGs, and then tallied the number of CBGs with one or more supplemental index exceeding the 80th percentile. This analysis was performed in QGIS.

## Results

Following these methods, we find that there are 35 proposed CCS projects, and 33 of them (94.3%) are located within three miles of an EJ community, while only two of them (5.7%) are not (Fig 1, Table 1). Of the CBGs falling in whole or partially within the 3-mile fenceline areas of an implicated coal plant, 497 meet the criteria for being considered an EJ CBG, while 188 do not (Table 2). These 497 CBGs are particularly vulnerable. A majority of them (85.1%) already face heightened burden, as indicated by having one or more supplemental EJSCREEN indices exceeding the 80th percentile.

## Discussion

Our findings indicate that the vast majority of proposed CCS facilities are located within three miles of an EJ community. Furthermore, the vast majority of the EJ CBGs within a 3-mile radius of these facilities face heightened environmental stress, meaning that the overwhelming majority of EJ CBGs are already overburdened to some degree. These results

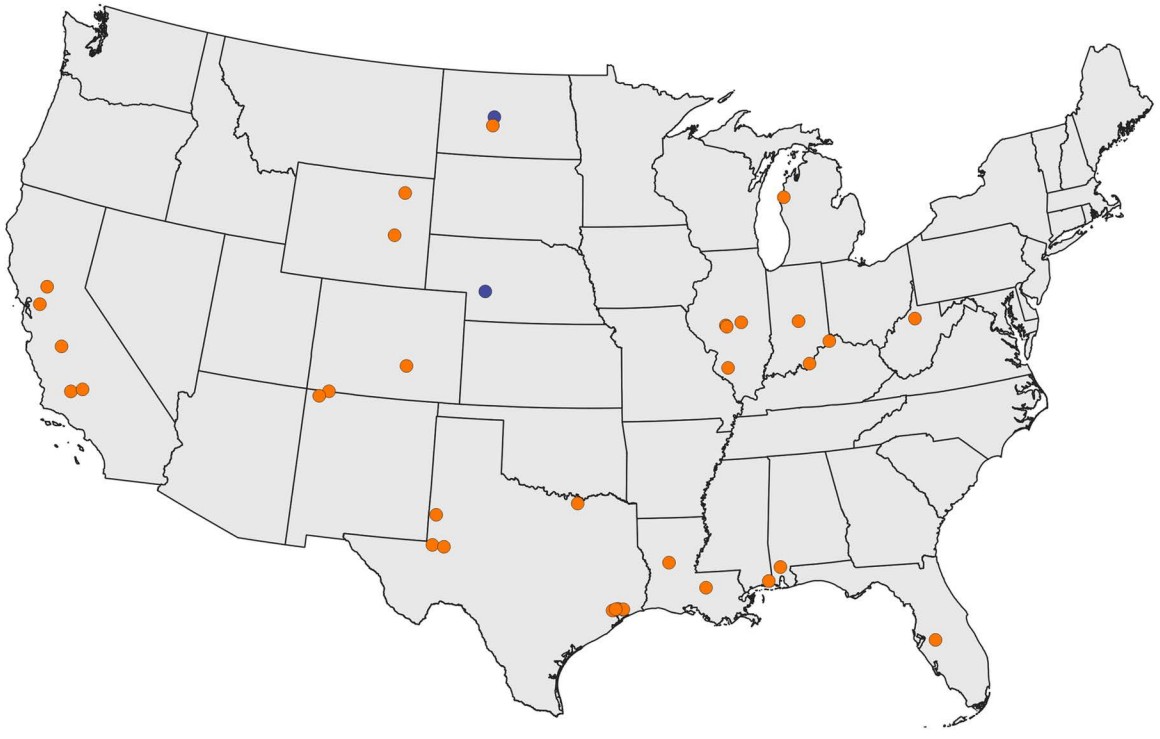

Orange = located within three miles of an EJ community, blue = not located within three miles of an EJ community

**Fig 1. Map of proposed CCS projects in the US power generation sector and co-location with EJ communities.**

**Table 1. Proposed CCS projects at power plants and co-location with an EJ community.**

|  | Number of projects (% of total) |
| --- | --- |
| Co-located with an EJ community | 33 (94.3%) |
| Not co-located with an EJ community | 2 (5.7%) |
| Total CCS projects proposed at power plants | 35 (100%) |

**Table 2. EJ Census Block Groups within three miles of a proposed CCS power plant project.**

|  | Number of EJ CBGs (% of total) |
| --- | --- |
| Facing heightened environmental stress | 423 (85.1%) |
| Not facing heightened environmental stress | 74 (14.9%) |
| Total EJ CBGs within three miles of CCS projects | 497 (100%) |

illustrate both the importance and feasibility of conducting analysis on the disproportionate impacts that EJ communities may face from CCS buildout in the US power sector. Our findings align with previous research that has shown that fossil fuel power plants are disproportionately sited in EJ communities [4–6]. The development of new CCS projects such as the ones assessed in this study can exacerbate the already disproportionate burden that EJ communities bear, building on a legacy of historical injustice. These findings are critical as to date, the federal government has not conducted any formal

EJ or cumulative impacts (CI) analysis of proposed CCS facilities in both regulatory and non-regulatory contexts. One prominent regulatory example is the EPA's classification of CCS as a BSER despite not assessing potential EJ impacts, including the changes to co-pollutant emission levels at any plants, or cumulative impacts of CCS for existing and new natural gas-fired power plants [9]. Outside of the regulatory sphere, the DOE has already advanced CCS demonstration projects at power plants despite a lack of research on the EJ or CI impacts of these facilities [31,32]. Even though billions of dollars in federal funding have already been allocated to these types of projects, it was only in 2024 that the National Academies of Sciences, Engineering, and Medicine (NASEM), a congressionally chartered organization that provides scientific expertise on timely national policy issues, launched a call for experts to assess the safety and EJ impacts of carbon management [33].

Our research builds on the nascent academic literature that acknowledges the impact of CCS on frontline communities and its potential to exacerbate existing injustices. One study that outlines the biophysical risks of CDR technologies, including CCS, establishes that frontline communities are most at risk from the impacts of CCS capture, transport, and storage [34]. Nielsen and colleagues apply a justice lens to review the literature on community perceptions of CCS projects, and find potential concerns related to both distributional and procedural justice [35]. One of their reviewed studies reported concerns about the inequitable distribution of risks from $CO_2$ leakage during storage [36]. Several of the reviewed studies reported instances of procedural injustices in which project information was not easily accessible, with some communities lacking the resources to engage with developers and negotiate with them, and lower-wealth communities feeling that they would be unable to influence or be protected from negative outcomes from CCS over a project's lifetime [37,38].

Similarly, McLaren et al. [39] conduct an overview of injustices along the CCS lifecycle, including the planning, capture, transport, and storage stages. They discuss the risks of enabling increased fossil fuel use, increased energy and water consumption during the capture phase, and $CO_2$ leakage during transport, storage, and decommissioning [39]. In a more recent review, Rojas-Rueda, McAuliffe, and Morales-Zamora [40] outline the health equity risks of CCS projects and their potential to amplify existing energy injustices, such as the exposure to environmental pollutants, risks during transport and storage, and disproportionate siting in EJ communities. Our study contributes empirical evidence to support these emerging claims about the potential injustices of CCS projects.

While the academic literature on the EJ impacts of CCS is nascent, substantial scientific research exists that points toward detrimental environmental and health impacts arising from the deployment of CCS at power plants. For example, at the point of capture, the energy penalty associated with powering the capture process may lead to increases in non-GHG emissions, such as mercury, particulate matter, sulfur dioxide, nitrogen oxides, and hazardous air pollutants, if the process is powered using fossil fuels [41]. Studies have reported energy penalties associated with carbon capture between 15 and 44 percent [42]. Another risk during the capture phase is the disposal of amine-based solvents used to separate $CO_2$ from the flue gas, which may present environmental risks (see, e.g., [43,44]). As another example, $CO_2$ transport via pipelines presents additional risks in the case of leakage or pipeline failure [45–47]. Finally storage of $CO_2$ can pose risks to local drinking water [48–50] and potentially contribute to seismic activity [51–53]. Major gaps remain in the literature on how each of these risks may differentially impact EJ communities, yet the co-location of potential CCS infrastructure with EJ communities, as we have begun to document here, points toward the urgent need to fill this gap.

## Limitations

A limitation of this study is that it does not capture the entire ecosystem of CCS projects in the power sector as it reflects only the projects included in the aforementioned databases. A forthcoming EPA power sector rule for existing natural gas plants may also impact where CCS projects are proposed, but the rule has not been finalized at the time of this writing and therefore its implications can not be captured in this analysis. While our definition of EJ is illustrative of a definition that EJ groups have used for other policy and research purposes, other definitions exist that could be used for future analyses. Moreover, as is the case in other EJ-focused spatial analyses that investigate "community" impacts, the CBG

spatial unit used in this analysis is an artificial boundary and not necessarily representative of actual EJ communities. As more CCS projects are proposed and developed, qualitative analysis that investigates in-situ impacts on EJ communities would ground-truth and build on these preliminary spatial analyses. Finally, the buildout of $CO_2$ storage and transportation infrastructure that would be needed to make these and other CCS projects possible was beyond the scope of this current analysis. Conducting an assessment of the co-location of proposed CCS storage facilities and pipelines with EJ communities could be an area of future research, allowing for a more complete picture of the full EJ costs and risks of implementing CCS.

Other critical limitations include the static nature of the analysis and potential biases introduced by the available data sources. Given the small sample size of the dataset, more in-depth statistical or spatial analyses such as regression models, spatial clustering techniques, or longitudinal comparisons were not used in this analysis, but would serve to add further nuance to understanding how carbon capture projects may disproportionately affect EJ communities. In addition to the quantitative assessment performed in this study, an analysis of qualitative insights from impacted communities would provide additional insight into the potential EJ impacts of CCS projects in the power sector and further strengthen these preliminary findings.

## Conclusion

The results of this co-location analysis highlight that proposed CCS facilities in the US power sector are disproportionately located in EJ communities. In reviewing the CCS riskscape, there is concern that these facilities may present additional risk to already overburdened EJ communities. To date, there is a scarcity of EJ or cumulative impacts analyses to evaluate the potential impacts of proposed CCS facilities. Future CCS initiatives could benefit from undergoing these types of analyses to ensure that projects will not exacerbate existing injustices and protect low-income communities and communities of color that already face the disproportionate impacts of energy infrastructure siting. Given that the majority of the proposed CCS facilities in the power sector considered in our analysis are located in EJ communities, and that such communities are already overburdened, our results point to the importance of a precautionary approach to CCS implementation and more rigorous analysis and consultation with communities before projects are implemented.

A rigorous EJ analysis should consider the impacts to co-pollutant emissions from CCS in addition to GHG emissions and incorporate an assessment of CI and potential health impacts at relevant spatial scales. For CCS projects that do move forward, plants should be in compliance with environmental regulations before they are permitted to add new CCS equipment. More rigorous monitoring at the site, including fenceline air quality monitoring, should also be required. In terms of consultation, potentially affected communities should receive comprehensive information about a project early and often, be provided with technical support to assess information, and have the option to influence the outcome of the project. In places that are already overburdened, meaningful engagement practices must be employed, following the framework of other EJ protections like state-level CI laws in New York and New Jersey. A critical component of meaningful engagement is the option for a community to deny a facility from moving forward with its development, which would advance justice aims while protecting EJ communities from the burdens of energy infrastructure siting.

## Supporting information

**S1 Table. CCS Databases.**
(PDF)

**S1 Appendix. List of proposed CCS facilities.**
(PDF)

**S1 File. CCS_Facilities_Planned_8.31.23.csv.**
(CSV)

## Acknowledgments

The authors would like to thank Dr. Nicky Sheats and Thomas Ikeda at the John S. Watson Institute for Urban Policy and Research at Kean University, Ansha Zaman at the Center for Earth, Energy and Democracy, and Brooke Helmick at the New Jersey Environmental Justice Alliance for the feedback they provided throughout the analysis.

## Author contributions

**Conceptualization:** Yukyan Lam, Ana Isabel Baptista.

**Data curation:** Yukyan Lam, Jennifer Ventrella, Juan David Rodriguez.

**Formal analysis:** Yukyan Lam, Jennifer Ventrella, Juan David Rodriguez.

**Investigation:** Yukyan Lam, Jennifer Ventrella, Juan David Rodriguez.

**Methodology:** Yukyan Lam, Jennifer Ventrella, Ana Isabel Baptista, Juan David Rodriguez.

**Project administration:** Yukyan Lam, Ana Isabel Baptista.

**Supervision:** Yukyan Lam, Ana Isabel Baptista.

**Visualization:** Yukyan Lam, Jennifer Ventrella, Juan David Rodriguez.

**Writing – original draft:** Yukyan Lam, Jennifer Ventrella.

**Writing – review & editing:** Yukyan Lam, Jennifer Ventrella, Ana Isabel Baptista.

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
