## [Decision Letter · Decision Letter 0]

30 Dec 2024

PONE-D-24-37993Analysis of proposed carbon capture projects in the US power sector and co-location with environmental justice communitiesPLOS ONE

Dear Dr. Lam,

Thank you for submitting your manuscript to PLOS ONE. After careful consideration, we feel that it has merit but does not fully meet PLOS ONE’s publication criteria as it currently stands. Therefore, we invite you to submit a revised version of the manuscript that addresses the points raised during the review process.

We look forward to receiving your revised manuscript.

Kind regards,

Diogo Guedes Vidal, PhD

Academic Editor

PLOS ONE

2. We note that your Data Availability Statement is currently as follows: [All relevant data are within the manuscript and its Supporting Information files.] Please confirm at this time whether or not your submission contains all raw data required to replicate the results of your study. Authors must share the “minimal data set” for their submission. PLOS defines the minimal data set to consist of the data required to replicate all study findings reported in the article, as well as related metadata and methods (https://journals.plos.org/plosone/s/data-availability#loc-minimal-data-set-definition).

3. We note you have included a table to which you do not refer in the text of your manuscript. Please ensure that you refer to Table 1 and 2 in your text; if accepted, production will need this reference to link the reader to the Table.

Additional Editor Comments (if provided):

Reviewers' comments:

Reviewer's Responses to Questions

**Comments to the Author**

1. Is the manuscript technically sound, and do the data support the conclusions?

Reviewer #1: Yes

Reviewer #2: No

2. Has the statistical analysis been performed appropriately and rigorously? 

Reviewer #1: Yes

Reviewer #2: No

3. Have the authors made all data underlying the findings in their manuscript fully available?

Reviewer #1: Yes

Reviewer #2: No

4. Is the manuscript presented in an intelligible fashion and written in standard English?

Reviewer #1: Yes

Reviewer #2: Yes

5. Review Comments to the Author

Reviewer #1: The manuscript is clear and brings an interesting analysis, that supports a problematic issue on environmental justice. In suggest minor adjustments that, in my view, will strengh the research:

1. Row 26: in the introduction, include EJ right after the first mention to Environmental Justice;

2. Rows 211-213, (..."federal government has not conducted any formal EJ or cumulative impacts analysis of proposed CCS facilities") contextualyse/explains better this "no action" (is it a regulory context?).

3. Rows 287-288: Elaborate on short suggestions on how more rigorous analysis and consultation with the population could be implemented. A short paragraph is enough.

Reviewer #2: The manuscript titled "Analysis of Proposed Carbon Capture Projects in the US Power Sector and Co-Location with Environmental Justice Communities" addresses an important topic at the intersection of climate change mitigation and environmental justice. However, the study's current form lacks sufficient detail and clarity, particularly regarding the methodology and the robustness of the analyses, which significantly undermines its potential contribution.

The methodological section requires substantial revision. The authors must specify how the carbon capture projects were selected. For example, the manuscript refers to the use of databases, but it fails to provide clear criteria for inclusion or exclusion of projects. Were the projects selected based on geographical distribution, scale, or potential environmental impact? Additionally, the process of estimating the proximity of these projects to environmental justice (EJ) communities is described superficially. It is critical to detail the mapping techniques used, the accuracy of geolocation data, and how overlapping boundaries or shared fenceline populations were handled during analysis.

The manuscript would also benefit from more rigorous analytical approaches. The findings rely heavily on basic counts and percentages, but fail to explore more in-depth statistical or spatial analyses that could provide greater insight into the relationships between carbon capture projects and environmental burdens. For example, employing regression models, spatial clustering techniques, or longitudinal comparisons could help strengthen the conclusions. These methodologies could provide a more nuanced understanding of how carbon capture projects disproportionately affect EJ communities.

Furthermore, the conclusions are broad and lack precision. The authors should use the findings to develop targeted policy recommendations or propose specific mitigation measures to address the inequities identified.

A significant shortcoming of the study is the lack of a detailed discussion on its limitations. The authors briefly mention data incompleteness but fail to explore other critical limitations, such as the static nature of the analysis, potential biases introduced by the choice of data sources, or the lack of qualitative insights from impacted communities.

In its current form, the manuscript does not meet the standards for publication. The authors are encouraged to resubmit the study after addressing these issues with a clearer methodological framework, more robust analyses, and precise conclusions.

6. PLOS authors have the option to publish the peer review history of their article (what does this mean?). If published, this will include your full peer review and any attached files.

Reviewer #1: No

Reviewer #2: **Yes: **Hélder Silva Lopes

---

## [Author Response · Author response to Decision Letter 1]

4 Mar 2025

Thank you, we have updated the manuscript to ensure it meets PLOS ONE’s style requirements.

2. We note that your Data Availability Statement is currently as follows: [All relevant data are within the manuscript and its Supporting Information files.] Please confirm at this time whether or not your submission contains all raw data required to replicate the results of your study. Authors must share the “minimal data set” for their submission. PLOS defines the minimal data set to consist of the data required to replicate all study findings reported in the article, as well as related metadata and methods (https://journals.plos.org/plosone/s/data-availability#loc-minimal-data-set-definition).

Thank you, we have uploaded the following data set as a Supporting Information File:

S1 File. CCS_Facilities_Planned_8.31.23.csv

We have provided the power plant data which is the minimal data set to replicate the study, since the other data are census data available and maintained by a third party (US government).

3. We note you have included a table to which you do not refer in the text of your manuscript. Please ensure that you refer to Table 1 and 2 in your text; if accepted, production will need this reference to link the reader to the Table.

Thank you, we have included the following reference in the text to link the reader to Table 1 and 2:

“Following these methods, we find that there are 35 proposed CCS projects, and 33 of them (94.3%) are located within three miles of an EJ community, while only two of them (5.7%) are not (Fig 1, Table 1). Of the census block groups falling in whole or partially within the 3-mile fenceline areas of an implicated coal plant, 497 meet the criteria for being considered an EJ census block group, while 188 do not (Table 2).” (p. 9, lines 183, 185)

Comments to the Author

Reviewer #1: The manuscript is clear and brings an interesting analysis, that supports a problematic issue on environmental justice. In suggest minor adjustments that, in my view, will strengh the research:

Thank you for your positive review of the paper and for your suggestions for improvement.

1. Row 26: in the introduction, include EJ right after the first mention to Environmental Justice;

Thank you, we have updated the sentence in the introduction to read: “The equity implications of deploying these technologies–particularly their impacts on low-income communities and communities of color, or environmental justice (EJ) communities–have been understudied.” (p. 2, line 29)

2. Rows 211-213, (..."federal government has not conducted any formal EJ or cumulative impacts analysis of proposed CCS facilities") contextualyse/explains better this "no action" (is it a regulory context?).

We appreciate the request for clarification, and have added additional context to the statement for clarification: “These findings are critical as to date, the federal government has not conducted any formal EJ or cumulative impacts (CI) analysis of proposed CCS facilities in both regulatory and non-regulatory contexts. One prominent regulatory example is the EPA’s classification of CCS as a BSER despite not assessing potential EJ impacts, including the changes to co-pollutant emission levels at any plants, or cumulative impacts of CCS for existing and new natural gas-fired power plants [9]. Outside of the regulatory sphere, the DOE has already advanced CCS demonstration projects at power plants despite a lack of research on the EJ or CI impacts of these facilities [31,32]. Even though billions of dollars in federal funding have already been allocated to these types of projects, it was only in 2024 that the The National Academies of Sciences, Engineering, and Medicine (NASEM), a congressionally chartered organization that provides scientific expertise on timely national policy issues, launch a call for experts to assess the safety and EJ impacts of carbon management [33]. (pp. 10-11, lines 210-219)

3. Rows 287-288: Elaborate on short suggestions on how more rigorous analysis and consultation with the population could be implemented. A short paragraph is enough.

We appreciate your feedback on elaborating on suggestions to improve the rigor of analysis and consultation for CCS. We have added the following: “A rigorous EJ analysis should consider the impacts to co-pollutant emissions from CCS in addition to GHG emissions and incorporate an assessment of cumulative impacts and potential health impacts at relevant spatial scales. For CCS projects that do move forward, plants should be in compliance with environmental regulations before it is permitted to add new CCS equipment. More rigorous monitoring at the site, including fenceline air quality monitoring, should also be required. In terms of consultation, potentially affected communities should receive comprehensive information about a project early and often, be provided with technical support to assess information, and have the option to influence the outcome of the project. In places that are already overburdened, meaningful engagement practices must be employed, following the framework of other EJ protections like state-level CI laws in New York and New Jersey. A critical component of meaningful engagement is the option for a community to deny a facility from moving forward with its development, which would advance justice aims while protecting EJ communities.” (p. 14, lines 294-306)

Reviewer #2: The manuscript titled "Analysis of Proposed Carbon Capture Projects in the US Power Sector and Co-Location with Environmental Justice Communities" addresses an important topic at the intersection of climate change mitigation and environmental justice. However, the study's current form lacks sufficient detail and clarity, particularly regarding the methodology and the robustness of the analyses, which significantly undermines its potential contribution.

Thank you, we appreciate your thoughtful review and recommendations to improve the detail, clarity, and rigor of the methodology and analysis.

The methodological section requires substantial revision. The authors must specify how the carbon capture projects were selected. For example, the manuscript refers to the use of databases, but it fails to provide clear criteria for inclusion or exclusion of projects. Were the projects selected based on geographical distribution, scale, or potential environmental impact?

Thank you for highlighting the omissions and lack of clarity in our methodology section. For additional context for the reviewer, the 35 projects analyzed are all of the proposed CCS projects in the power sector across the databases we used, which to our knowledge are the leading publicly available industry and government databases of CCS projects. Given the small sample size and that we included all of the projects we found, we could not conduct more in-depth spatial or statistical analysis with our dataset. The project scope and methodology were co-developed with environmental justice leaders and representatives of impacted communities to better understand the potential impacts of proposed projects. Although the existing dataset is small, the potential for CCS deployment is much bigger because significant investments have already been made to scale up this technology.

Given this context, we have added the following paragraph to clarify and address missing information:

“For projects not included in the DOE database, we estimated their location based on publicly available data about the power plant at which they would be installed or at an approximate location. Our inclusion criteria included any planned CCS facility in the power sector across all four databases. The application of this criteria resulted in a final list of 35 planned facilities included in the analysis can be found in S1 Appendix. We did not include operational plants because at the time of the analysis the only operational CCS facilities in the power sector were at pilot/demo scale, not commercial scale.” (p. 5, lines 107-108)

Additionally, the process of estimating the proximity of these projects to environmental justice (EJ) communities is described superficially. It is critical to detail the mapping techniques used, the accuracy of geolocation data, and how overlapping boundaries or shared fenceline populations were handled during analysis.

Thank you for pointing out the further clarification needed on the process of estimating the proximity of these projects to EJ communities. We have included the following text to outline our mapping techniques:

“The proximity analysis was conducted in QGIS (version 3.32) using the 'Buffer' tool to create 3-mile radius buffer zones around the points of interest. Input data included a point layer representing the facility locations, a shapefile layer representing census block groups, and a csv layer that included socio-demographic data. The coordinate reference system (CRS) was set to ESRI 102003 to ensure accurate distance calculations in miles. Features within the buffer zones were identified and analyzed through spatial joins with the EJ CBG and supplemental indices datasets. Additional documentation for each tool can be found at the QGIS website.” (p. 7, lines 150-156)

To provide further detail on the accuracy of geolocation data, we have added the following detail to our existing documentation:

“Latitude and longitude coordinates were provided for projects appearing in the DOE database. The database does not document how it obtained the coordinates, and we therefore cannot verify the accuracy of these coordinates. For projects not included in the DOE database, we estimated their location based on publicly available data about the power plant at which they would be installed or at an approximate location. Approximate locations were estimated for seven of the 35 plants. To perform the estimation, secondary research sources such as project announcements, reports, or other documentation were used to triangulate an approximate project location. Approximated locations are a potential source of error in the analysis, however, these are random, not systematic, errors, and therefore should not inherently bias the data towards or away from any particular population.” (p. 5, lines 102-107)

The description of how overlapping boundaries were handled during the analysis is shared here:

“In order to ensure that CBGs located near more than one implicated facility were not double-counted, we merged three-mile buffers around the implicated facilities, intersected the buffers with the EJ CBGs, and then tallied the number of block groups with one or more supplemental index exceeding the 80th percentile.” (p. 8, lines 175-178)

The manuscript would also benefit from more rigorous analytical approaches. The findings rely heavily on basic counts and percentages, but fail to explore more in-depth statistical or spatial analyses that could provide greater insight into the relationships between carbon capture projects and environmental burdens. For example, employing regression models, spatial clustering techniques, or longitudinal comparisons could help strengthen the conclusions. These methodologies could provide a more nuanced understanding of how carbon capture projects disproportionately affect EJ communities.

We very much appreciate these suggestions, which we agree would be useful analyses to provide additional insight into the relationships between carbon capture projects and environmental burdens, but are beyond the scope of this paper. As described above, we do not have a large enough sample size to perform these more in-depth analyses. We have also added a point to address this in our limitations section, detailed below. This is a common challenge for environmental justice studies, as these are preliminary, emerging, and often under-scrutinized threats to EJ communities’ wellbeing and therefore often entail small datasets or missing data, a lack of geographically precise data, and/or a lack of statistical precision. The goal of our study is to unearth a problem that is considered relatively invisible from a policy and scientific perspective and we hope that the reviewer still sees the value in this study.

Furthermore, the conclusions are broad and lack precision. The authors should use the findings to develop targeted policy recommendations or propose specific mitigation measures to address the inequities identified.

Thank you for highlighting this gap in the conclusion. We have added new content that includes more targeted policy recommendations and specific mitigation measures to address the inequities identified: “A rigorous EJ analysis should consider the impacts to co-pollutant emissions from CCS in addition to GHG emissions and incorporate an assessment of cumulative impacts and potential health impacts at relevant spatial scales. For CCS projects that do move forward, plants should be in compliance with environmental regulations before it is permitted to add new CCS equipment. More rigorous monitoring at the site, including fenceline air quality monitoring, should also be required. In terms of consultation, potentially affected communities should receive comprehensive information about a project early and often, be provided with technical support to assess information, and have the option to influence the outcome of the project. In places that are already overburdened, meaningful engagement practices must be employed, following the framework of other EJ protections like state-level CI laws in New York and New Jersey. A critical component of meaningful engagement is including the option for a community to deny a facility from moving forward with its development, which would advance justice aims while protecting EJ communities from the burdens of energy infrastructure siting.” (p. 14, lines 294-306)

A significant shortcoming of the study is the lack of a detailed discussion on its limitations. The authors briefly mention data incompleteness but fail to explore other critical limitations, such as the static nature of the analysis, potential biases introduced by the choice of data sources, or the lack of qualitative insights from impacted communities.

Thank you for highlighting this shortcoming. We have added the following statement to address additional study limitations: “Other critical limitations include the static nature of the analysis and potential biases introduced by the available data sources. Given the small sample size of the dataset, more in-depth statistical or spatial analyses such as regression models, spatial clustering techniques, or longitudinal comparisons were not used in this analysis, but would serve to add further nuance to understanding how ca

---

## [Decision Letter · Decision Letter 1]

16 Apr 2025

Analysis of proposed carbon capture projects in the US power sector and co-location with environmental justice communities

PONE-D-24-37993R1

Dear Dr. Lam,

We’re pleased to inform you that your manuscript has been judged scientifically suitable for publication and will be formally accepted for publication once it meets all outstanding technical requirements.

Kind regards,

Diogo Guedes Vidal, PhD

Academic Editor

PLOS ONE

Additional Editor Comments (optional):

Reviewers' comments:

Reviewer's Responses to Questions

**Comments to the Author**

1. If the authors have adequately addressed your comments raised in a previous round of review and you feel that this manuscript is now acceptable for publication, you may indicate that here to bypass the “Comments to the Author” section, enter your conflict of interest statement in the “Confidential to Editor” section, and submit your "Accept" recommendation.

Reviewer #1: All comments have been addressed

Reviewer #2: All comments have been addressed

2. Is the manuscript technically sound, and do the data support the conclusions?

Reviewer #1: Yes

Reviewer #2: Yes

3. Has the statistical analysis been performed appropriately and rigorously? 

Reviewer #1: Yes

Reviewer #2: Yes

4. Have the authors made all data underlying the findings in their manuscript fully available?

Reviewer #1: Yes

Reviewer #2: Yes

5. Is the manuscript presented in an intelligible fashion and written in standard English?

Reviewer #1: Yes

Reviewer #2: Yes

6. Review Comments to the Author

Reviewer #1: (No Response)

Reviewer #2: After the revisions made, the paper can be accepted. All comment were addressed.

Congratulations for the work.

7. PLOS authors have the option to publish the peer review history of their article (what does this mean?). If published, this will include your full peer review and any attached files.

Reviewer #1: No

Reviewer #2: No

---

## [Editor Report · Acceptance letter]

PONE-D-24-37993R1

PLOS ONE

Dear Dr. Lam,

I'm pleased to inform you that your manuscript has been deemed suitable for publication in PLOS ONE. Congratulations! Your manuscript is now being handed over to our production team.

Kind regards,

on behalf of

Dr. Diogo Guedes Vidal

Academic Editor

PLOS ONE